# A Unique Case of Bilateral Thalamic High-Grade Glioma in a Pediatric Patient with LI-Fraumeni Syndrome: Case Presentation and Review of the Literature

**Raffaella Messina** [1,*,†] , **Gerardo Cazzato** [2,†] , **Teresa Perillo** [3] , **Vita Stagno** [4] , **Valeria Blè** [1] , **Mariachiara Resta** [5] , **Francesco De Leonardis** [3] , **Nicola Santoro** [3] , **Francesco Signorelli** [1] and **Giuseppe Ingravallo** [2,*]

1   Division of Neurosurgery, Department of Basic Medical Sciences, Neurosciences and Sense Organs, University "Aldo Moro" of Bari, 70124 Bari, Italy; valeryblu13@gmail.com (V.B.); francesco.signorelli@uniba.it (F.S.)
2   Department of Emergency and Organ Transplantation—Section of Pathology, University "Aldo Moro" of Bari, 70124 Bari, Italy; gerycazzato@hotmail.it
3   Department of Pediatric Oncology and Hematology, University "Aldo Moro" of Bari, 70124 Bari, Italy; terryperillo@hotmail.com (T.P.); fdl111@libero.it (F.D.L.); nico.santoro1956@libero.it (N.S.)
4   Department of Neurosurgery North Bristol Trust, Southmead Hospital, Bristol BS10 5NB, UK; vitastagno@gmail.com
5   Division of Neuroradiology, Department of Basic Medical Sciences, Neurosciences and Sense Organs, University "Aldo Moro" of Bari, 70124 Bari, Italy; mariachiararesta@yahoo.it
*   Correspondence: raffaella.messina@uniba.it (R.M.); giuseppe.ingravallo@uniba.it (G.I.)
†   These authors contributed equally to this work.

**Abstract:** Li-Fraumeni syndrome (LFS) is a rare high-penetrance and autosomal-dominant pathological condition caused by the germline mutation of the TP53 gene, predisposing to the development of tumors from pediatric age. We conducted a qualitative systematic review following the ENTREQ (Enhancing Transparency in Reporting the Synthesis of Qualitative Research) framework. A search was made in MEDLINE/Pubmed and MeSH Database using the terms "Li-Fraumeni" AND "pediatric high-grade glioma (HGG)", identifying six cases of HGGs in pediatric patients with LFS. We added a further case with peculiar features such as no familiar history of LFS, association of embryonal rhabdomyosarcoma and bithalamic HGG, whose immunohistochemical profile was accurately defined by Next Generation Sequencing. Knowledge synthesis and case analysis grounded the discussion about challenges in the management of this pathology in pediatric age.

**Keywords:** Li-Fraumeni syndrome; embryonal rhabdomyosarcoma; high-grade bithalamic glioma; NTRK genes; p53 protein

## 1. Introduction

Li-Fraumeni syndrome (LFS) predisposes to the early onset of a wide variety of cancers associated to a germ line mutation in the TP53 gene, located on the 17p13.1 chromosome. This mutation has a high-penetrance and an autosomal-dominant transmission pattern. The p53 protein encoded by the homologous gene plays a complex and important role in repairing DNA damage and preventing cellular oxidative stress, acting as a tumor suppressor factor. Therefore, mutations causing a loss of its function, as happens in LFS, predispose to the onset of tumors such as sarcomas, osteosarcomas, breast cancers and pheochromocytomas from childhood. Among brain tumors, the most frequently found are low-grade gliomas, medulloblastomas and choroid plexus carcinomas in pediatric age and high-grade gliomas (HGGs) in adulthood. Some patients develop multiple tumors from pediatric age.

*Case Presentation*

A previously healthy boy, born of full-term normal pregnancy, at the age of 18 months developed a swelling of the left thigh at the level of the femoral rectum muscle. Family medical history was relevant for ovarian cancer in the maternal great-grandmother and leukemia in his father's cousin, 52 years old. He underwent an incisional biopsy of the lesion. The tissue was fixed in formalin, sampled and embedded in paraffin, sectioned at 4 μm and stained with haematoxylin-eosin. Histologically the neoplasm was characterized by a proliferation of fused cells and isolated large cellular elements, with large eosinophilic cytoplasm and voluminous and dysmorphic nuclei. The tumor cells showed consistent immunoreactivity for desmin and focally for smooth-muscle actin and actin HHF35, compatible with embrional rhabdomyosarcoma with fused and anaplastic cells (Figure 1). Molecular analysis was positive for MyoD1 and negative for PAX3-FOXO1, PAX7-FOXO1, PAX3-NCOA1, SRF4-NCOA1. A brain MRI performed at this stage did not show any anomaly.

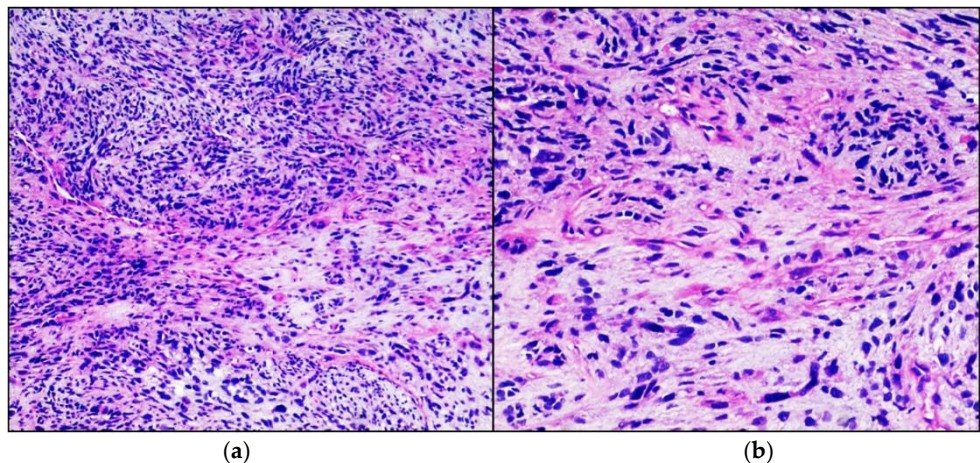

(a) (b)

**Figure 1.** Embrional Rhabdomyosarcoma. Hematoxylin-eosin stained photomicrographs, with original magnification 100× (**a**) and 200× (**b**), showing characteristic intersecting fascicle of spindle cells with eosinophilic cytoplasm and iperchromic pleomorphic nuclei.

The child underwent neoadjuvant polychemotherapy, according to the international treatment protocol EPSSGRMS 2005, and after nine months had a wide surgical resection of the mass infiltrating the muscles of the left anterior loggia thigh. Post-operative magnetic resonance imaging (MRI) showed complete tumor removal with radiological remission at 18-month follow-up.

Twenty months later the child started to experience intentional and resting tremor affecting mainly the upper right limb. A brain and spinal MRI showed a bithalamic bicaudate signal abnormality suggestive of malignancy (Figure 2a,b). Multidisciplinary team discussion recommended surgical biopsy in order to obtain histological and immunohistochemical subtyping. The stereotactic biopsy target was identified on the MRI spectroscopy at the level of the inferior portion of the left thalamus, as this was the site of choline peak with low M-acetylaspartate level, expected to have greater tumor activity (Figure 2c,d).

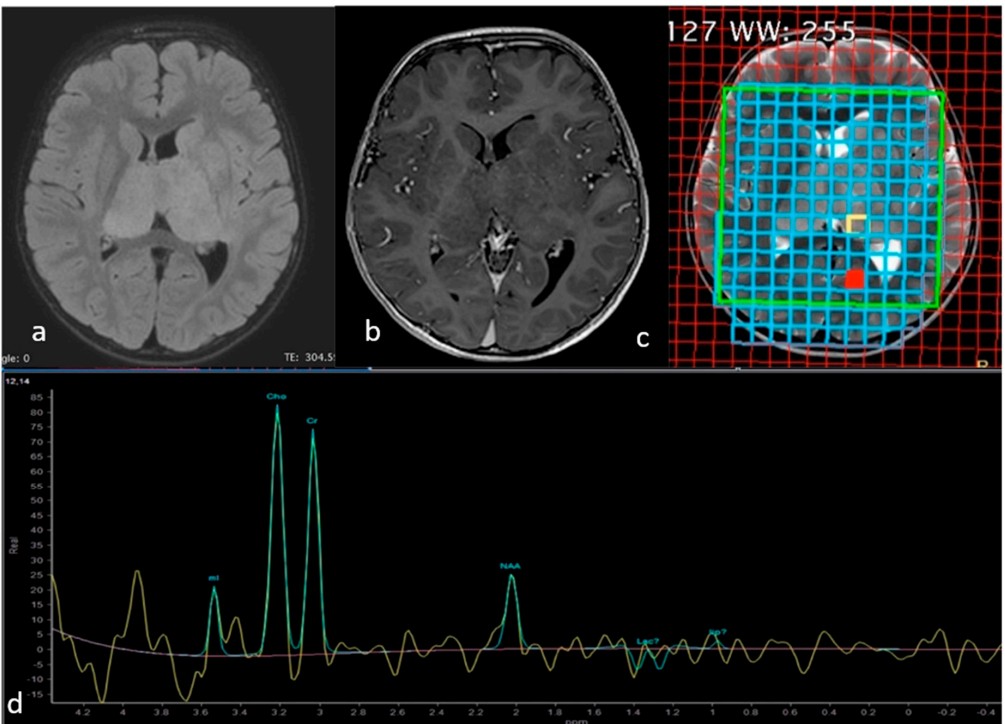

**Figure 2.** Axial FLAIR (**a**), Gadolinium-enhanced T1-weighted (**b**) and T2-weighted (**c**) MR images showing the lesion involving both thalami and head of caudate nuclei with no evident contrast enhancement. The spectrum (**d**) at the level of the yellow square in the left pulvinar corresponded to the area of highest cholin peak, where NAA was markedly decreased, chosen as the biopsy target.

Intraoperative frozen section on stereotactic biopsy confirmed a high-grade glial lesion, in agreement with final histological and immunohistochemical diagnosis of high-grade midline glioma (Figure 3). Immunohistochemical investigations tested positive diffusely for acid gliofibrillar protein (GFAP) and focally for p53 protein; reactions for B-RAF, desmin and IDH-1 were negative. The H3.3K27M protein was not expressed. Fusion-transcript of NTRK1 gene was absent, while it was present for NTRK2 gene.

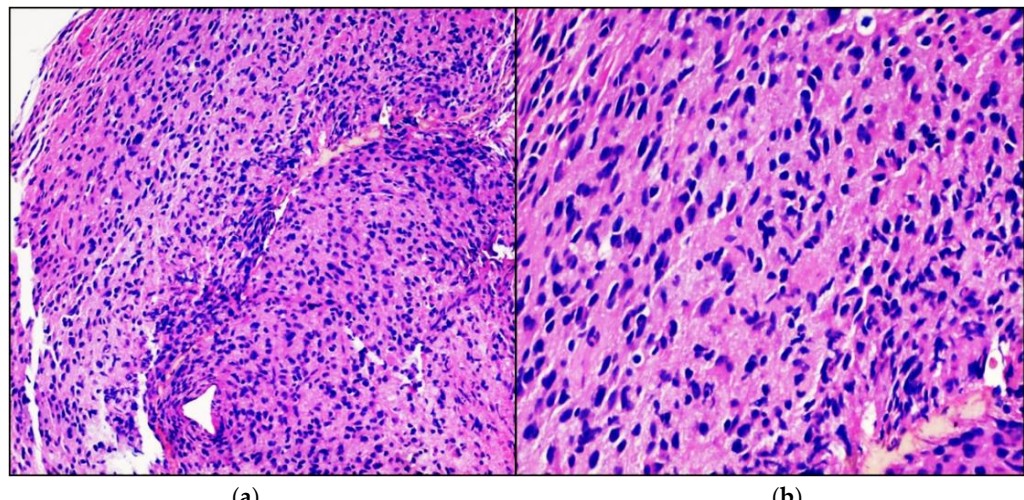

(**a**)　　　　　　　　　　　　　　　　　　　　　　　(**b**)

**Figure 3.** High-grade glioma. Hematoxylin-eosin-stained photomicrographs with original magnification 100× (**a**) and 200× (**b**) showing diffusely infiltrating cigar shaped neoplastic astrocytes with obvious nuclear atypia.

On the strong suspicion of a genetically determined syndromic condition, the child was subjected to genetic investigation in search of mutations predisposing to cancer. DNA sequencing based on the canonical transcript NM_000546.5 showed an heterozygous mutation of exon 7 of TP53 gene: c.743G > A (p.Arg248Gln), confirming the diagnosis of LFS.

The patient underwent 17 sessions of cranial radiotherapy at the dose of 32 Gray in daily fractions of 1.8 Grays, along with a rehabilitation program and neuropsychiatric support therapy. One-month follow-up MRI showed no significant tumor modification. However, the child died two months after the end of radiotherapy due to rapid deterioration of vital functions. Total survival following the diagnostic stereotactic biopsy was four months.

## 2. Methods

We conducted a qualitative systematic review following the ENTREQ (Enhancing Transparency in Reporting the Synthesis of Qualitative Research) framework [1]. This methodology was chosen as it better describes qualitative research according to the guidelines for reporting systemic reviews at: http://www.cochrane.de/de/LeitlinienForschungsberichte (accessed at 1 January 2021). A case-based literature search of pediatric cases of HGGs in the context of LFS was performed. The main search was conducted through PubMed and MeSH Database and the terms used were "Li-Fraumeni syndrome" and "pediatric gliomas". Only full text-paper (case series, reviews and research studies) published between January 2000 and November 2020 in peer-reviewed journals were included in the study. A systematic approach to the collected data and a collegial discussion between the authors led to a final version of the manuscript.

## 3. Results

We identified a total of 32 records through database searching according to our selection criteria (full-text only articles published from January 2000 to November 2020). Twenty-one publications were obtained through MEDLINE/PubMed and 11 from MeSH Database. Eleven studies were duplicate and were excluded from our review. A total of 21 peer-reviewed papers were screened based on this review's focus (HGG in pediatric patients with LFS). Eighteen papers were found not relevant and were excluded, leaving only three studies to be included in our qualitative analysis (Table S1). A total of six cases of pediatric patients with LFS and brain HGGs were identified (Table 1). Among these, four were described by Sloan and colleagues [2], who performed a comprehensive genomic characterization and studied the clinicopathologic features. García-Cárdenas et al. [3] reported a case of an Ecuadorian pediatric patient with anaplastic astrocytoma genetically characterized using a panel-based next-generation sequencing, whereas Zureick et al. [4] reported the case of a 14-year-old boy with LFS and a gigantocellular-type glioblastoma multiforme successfully treated with surgical resection, radiotherapy and Everolimus administration. In this case, target therapy was motivated by the results of whole exome and transcriptome sequencing revealing germline TP53 and somatic TSC2 mutations that emerged as markers of response to mTOR inhibitors [5].

Median age at tumor diagnosis was eight years. Among patients whose cancer history was available (five out of six), four had no previous history of other malignancies and one patient had previously been diagnosed with an osteosarcoma. Screening for tumor location disclosed five cases involving cerebral hemispheres and one involving thalamus. With regards to survival data, three patients with glioblastoma died and the other glioblastoma patient was still alive but with evidence of progressive disease after four months of follow up. Of the two patients diagnosed with anaplastic astrocytoma, one died after 17 months and the other one was alive with no evidence of progression at eight months follow up.

**Table 1.** High grade gliomas in patients with Li-Fraumeni syndrome: reported cases in the literature.

| Reference | No. | IDH Mutation Status | Age and Sex | Other Cancers | Family History | Brain Tumor Histology | Location | Follow Up |
|---|---|---|---|---|---|---|---|---|
| Sloan et al. [2] | 1 | Wildtype | 4, Male | None | Negative | Glioblastoma | Thalamus | 2 months |
| | 2 | Wildtype | 6, Male | None | Negative | Glioblastoma | Cerebral hemisphere | 12 months |
| | 3 | Wildtype | 11, Male | Osteosarcoma | Brain cancer (aunt); ovarian cancer (grandmother); rhabdomyosarcoma (uncle) | Anaplastic astrocytoma | Cerebral hemisphere | 8 months |
| | 4 | Wildtype | 6, Male | None | Negative | Glioblastoma | Cerebral hemisphere | 4 months |
| García-Cárdenas et al. [3] | 5 | n.e * | 13, Female | None | Breast cancer (mother and maternal grandmother); brain cancer (two maternal uncles and two maternal cousins) | Anaplastic astrocytoma | Cerebral hemisphere | 17 months |
| Zureick et al. [4] | 6 | n.a. ** | 14, Male | n.a. ** | n.a. ** | Glioblastoma | Cerebral hemisphere | 33 months |
| Our case | 7 | Wildtype | 3, Male | Rhabdomyosarcoma | Ovarian cancer (great-grandmother); leukemia (cousin) | Glioblastoma | Thalamus bilateral | 4 months |

* n.e. not evaluated; ** n.a. not available.

## 4. Discussion

It is estimated that about 10% of children with central nervous system (CNS) tumors harbor an underlying cancer-predisposition syndrome [6–9], the most frequent being LFS, whose prevalence is 1:5000–20,000. The diagnosis of LFS is established in a proband meeting all three classic criteria and/or tested positive to TP53 germline mutation. Classic LFS criteria include: (1) a sarcoma diagnosed before age 45 years; (2) a first-degree relative with any cancer diagnosed before age 45 years; (3) a first- or second-degree relative with any cancer diagnosed before age 45 years or a sarcoma diagnosed at any age [10]. The "Chompret's Revised Criteria" integrated the classic criteria as regards to the indication of testing for LFS [11].

In affected individuals, the risk of developing at least one tumor in lifetime is 75% among men and 93–100% among women. According to Olivier et al. [12] median age of onset of brain tumors in Li-Fraumeni syndrome is sixteen years and about 40% of affected children will develop a cancer by the age of 18 [13]. Pediatric CNS tumors in LFS represent the second most common cancer after adrenocortical carcinomas. Among brain tumors, the most frequent in pediatric age are low-grade gliomas, medulloblastomas and carcinomas of the choroid plexus, while adults have a higher incidence of HGG [6]. Thalamic HGG account for 13% of HGG in pediatric age and have a very unfavorable prognosis [14]. Our case is unique in several aspects: from an epidemiological point of view, to our knowledge there is no previous report of a bithalamic HGG in a pediatric patient with LFS. Despite patients with LFS being likely to develop a second malignancy before 18 years of age, there are no other reports in the literature of an embryonal rhabdomyosarcoma and an HGG in the same patient at such a young age. Our patient's family history did not reveal any criteria portending the diagnosis of LFS when the rhabdomyosarcoma was discovered, thus the child did not undergo genetic screening. However, according to the new "Chompret's Revised Criteria", LFS should have been suspected on the basis of the embryonal anaplastic subtype of the rhabdomyosarcoma and a brain MRI periodic screening would have possibly yielded to an earlier diagnosis of CNS involvement. Even though our patient's outcome likely would not have changed, our case points out the importance of genetic counselling when any of the new "Revised Chompret Criteria" are matched.

In our case, therapeutic strategy was discussed at the oncology multidisciplinary meeting. From a surgical point of view, the tumor was considered not amenable to gross total resection, since it was located in a highly functional area, multifocal and deep sited [15,16]. Instead, a stereotactic biopsy was recommended to facilitate histological diagnosis. Various stereotactic techniques have been described to obtain a thalamic biopsy. They can be frame-based, frameless or robot-assisted [17,18] and are usually performed through a trans frontal or trans cerebellar approach, but neither of them has shown significant reduced morbidity [19,20].

Despite the presence in our case of a somatic TP53 mutation, that in pediatric brain tumors is restricted to HGG and is independently associated with an adverse outcome [21], the patient underwent RT, considering the unavailability of other viable therapeutic alternatives. In LFS the negative prognostic factor of somatic TP53 mutation, associated with a TP53 germline mutation, hallmark of the disease, seems confirmed by some initial evidences: Boyle et al. [22] studied the effect of RT in fibroblasts derived from non-LFS, LFS (mutant TP53) and LFS (wild-type TP53) patients and observed that >50% chromosomal aberrations were accumulated in fibroblasts with mutant TP53, significantly more than in fibroblasts of the other groups, after irradiation. Moreover, TP53 mutated gliomas in LFS seem to have a higher potential for progression toward higher grades compared to TP53 wild-type gliomas [23]. Mutation of TP53, as occurs in LFS, not only neutralizes the P53 protein's normal tumor suppressive function, but may also disrupt cellular regulatory networks, thus enabling tumor cells to avoid genotoxic signals such as from gamma radiation, circumventing senescence and programmed cell death [21,24–26].

In case of LFS progression, ongoing studies are evaluating the therapeutic efficacy of Larotrectinib for those patients that, unlike our patient, have fusion-transcripts of the

NTRK1 gene [27–29]. The NTRK1 gene encodes a protein-kinase located on the cell surface, in particular on sensory neurons, that phosphorylates specific loci of other proteins, thus modulating their activity in order to regulate cell growth and survival. Mutations of this gene, especially rearrangements with other genes with fusion-transcripts formation, are related to various types of cancer and tumor-predisposing syndromes. An analysis has therefore been carried out using NGS (Next Generation Sequencing) technique on the glioma cells of our patient, showing the presence of fusion-transcripts of the NTRK2 gene. However, considering that the biological role of NTRK2 gene is still unknown and no indication for Larotrectinib use has been registered yet in this rare cohort of patients [30], no further treatment was administered. Therefore, further research on NTRK genes is needed, in order to define more target therapies indications and strategies [31,32].

## 5. Conclusions

HGG associated to LFS in the pediatric population are a rare entity and their survival remains poor, especially when surgery is not indicated, as in our case. Further efforts in genetic research are warranted to investigate the natural history of HGG in LFS to implement effective multimodal treatment approaches. In addition, to our knowledge this is the first case report of TP53 and NTRK2 genic co-alteration in a childhood glioma. For these various reasons, our work has the dual-purpose of sharing a unique case of bithalamic localization of an HGG in a LFS patient and of stimulating scientific debate about the best therapeutic approach to be used in such challenging cases.

**Supplementary Materials:** The following are available online at https://www.mdpi.com/article/10.3390/neurolint13020017/s1, Table S1: Flowchart of search strategy used in the study PRISMA 2009 Flow Diagram.

**Author Contributions:** R.M., T.P. and V.S. reviewed the literature, screened the abstracts of the reference list, deleted duplicates and citations not meeting the inclusion criteria, and assessed the articles; G.C. and V.B. wrote the case presentation; G.I. and G.C. performed the pathologic findings and the genetic analysis; M.R. elaborated the images and prepared the captions; R.M. and T.P. wrote the manuscript; R.M and V.S. prepared the review in line with the ENTREQ Statement using a 21-point checklist. The first version of the article was submitted to F.D.L. and N.S., who contributed to perform the discussion; F.S. critically revised the manuscript for intellectual content. Through an iterative approach and an analysis of the pertinent literature, the corrected version of the article was discussed collegially and a final version was produced. All authors have read and agreed to the published version of the manuscript.

**Funding:** This research received no external funding.

**Institutional Review Board Statement:** Not applicable.

**Informed Consent Statement:** Informed consent was obtained from all subjects involved in the study.

**Data Availability Statement:** Not applicable.

**Acknowledgments:** The Authors thank Felice Giangaspero and Maura Massimino, who supported us in treatment strategy decisions.

**Conflicts of Interest:** The authors declare no conflict of interest.

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
