# Peer review of "A Unique Case of Bilateral Thalamic High-Grade Glioma in a Pediatric Patient with LI-Fraumeni Syndrome: Case Presentation and Review of the Literature"

_2035-8377, doi:10.3390/neurolint13020017_

Round 1
Reviewer 1 Report
Messina et al. present a rare midline high grade glioma in a paediatric patient with Li-Fraumeni syndrome. The case presentation is well written and sound. It fairly describes the clinical reasoning and the difficult choice of treatment of an extensive bi-thalamic high grade glioma in a syndrome. Best treatment modalities remain largely unknown. In this respect the work adds value to clinicians facing similar dilemma in the future. An interesting point raised in the discussion and common to all onco-suppressor genes diseases causes debate: should radiotherapy be administered ? Traditional thought has claimed a higher risk of in-field secondary malignancies after radiotherapy whereas other recent works show no difference, suggesting more a local recurrence rather than a new radiation-induced tumor. Effective management may lie in a targeted-therapy associated with conventional therapy.
Author Response
Dear Reviewer,
We appreciate your comments and we agree with you.
Best Regards
Raffaella Messina
Giuseppe Ingravallo
On behalf of the co-authors
Reviewer 2 Report
the authors describe a case of RMS and subsequent HGG occurring in the context of gremlin TP53 syndrome, and perform a qualitative systematic review of the literature on this important topic.
the order how authors present first the literature review and only thereafter the case report it is unusual and probably not particularly effective.
moreover, the description of the case is too long and focusing on aspects which are not particularly innovative e.g. biopsy technique, or particularly relevant for the overall case discussion.
I would suggest to try to outline the possible association between TP53 and NTRK fusion found, which to my personal opinion render the case more unique rather than the location and the TP53 alone.
not sure that a treatment with NTRKi was not indicated, as there is some evidence of efficacy irrespectively from the type of NTRK fusion, albeit those with NTRK2 might respond less.
in conclusion this case and the literature review summarise a well known association between TP53 and malignant glioma, which is common at somatic level, but less frequently encountered at the gremline.
the use of the revised TP53 screening criteria is very useful to raise awareness on this under diagnosed condition.
I think the case is worth for publication with some review as suggested above.
Author Response
Dear Reviewer,
We appreciate your helpful comments; we believe that our manuscript has been improved greatly through the incorporation of your suggestions.
Best Regards
Raffaella Messina
Giuseppe Ingravallo
On behalf of the co-authors
Response to Reviewer 2 Comments
Point 1: The authors describe a case of RMS and subsequent HGG occurring in the context of gremlin TP53 syndrome, and perform a qualitative systematic review of the literature on this important topic.
The order how authors present first the literature review and only thereafter the case report it is unusual and probably not particularly effective.
Response 1: The suggested correction has been made. The Case presentation has been brought forward compared to the literature review.
Point 2: Moreover, the description of the case is too long and focusing on aspects which are not particularly innovative e.g. biopsy technique, or particularly relevant for the overall case discussion.
Response 2: the description of biopsy technique has been removed, as suggested; in addition, the Fig. 3 has been deleted.
Point 3: I would suggest to try to outline the possible association between TP53 and NTRK fusion found, which to my personal opinion render the case more unique rather than the location and the TP53 alone.
Response 3: The conclusions have been partially re-written, incorporating this suggestion: “In addition, to our knowledge this is the first case report of TP53 and NTRK2 genic co-alteration in a childhood glioma” (line 225-226). Two references have been added [31- 32].
Point 4: Not sure that a treatment with NTRKi was not indicated, as there is some evidence of efficacy irrespectively from the type of NTRK fusion, albeit those with NTRK2 might respond less.
Response 4: Unfortunately, we couldn’t treat with NTRKi because no indication for Larotrectinib use were registered yet in this rare cohort of patients. In that time, in our country there were not available appropriate paediatric clinic trials.
The whole paper has been checked for typos and syntax mistakes.
In conclusion this case and the literature review summarize a well-known association between TP53 and malignant glioma, which is common at somatic level, but less frequently encountered at the germ-line.
The use of the revised TP53 screening criteria is very useful to raise awareness on this under diagnosed condition.
I think the case is worth for publication with some review as suggested above.
Round 2
Reviewer 2 Report
I am ok with the new draft submitted by Authors